# Pharmacokinetics and Safety of Doripenem in Healthy Chinese Subjects and Monte Carlo Dosing Simulations

**DOI:** 10.3390/antibiotics11070958

**Published:** 2022-07-16

**Authors:** Yu Wang, Xiaofen Liu, Kun Li, Yaxin Fan, Jicheng Yu, Hailan Wu, Yi Li, Xiaojie Wu, Beining Guo, Xin Li, Jiali Hu, Jufang Wu, Guoying Cao, Jing Zhang

**Affiliations:** 1Institute of Antibiotics, Huashan Hospital, Fudan University, Shanghai 200040, China; 20111220007@fudan.edu.cn (Y.W.); liuxiaofen@huashan.org.cn (X.L.); fanyaxin@fudan.edu.cn (Y.F.); wuhailan@huashan.org.cn (H.W.); 12220220011@fudan.edu.cn (Y.L.); guobeining@huashan.org.cn (B.G.); 18111220016@fudan.edu.cn (X.L.); hujiali@huashan.org.cn (J.H.); 2Key Laboratory of Clinical Pharmacology of Antibiotics, National Health Commission of the People’s Republic of China, Shanghai 200040, China; yujicheng@fudan.edu.cn (J.Y.); wuxiaojie@fudan.edu.cn (X.W.); wujufang@huashan.org.cn (J.W.); 3National Clinical Research Center for Aging and Medicine, Huashan Hospital, Fudan University, Shanghai 200040, China; 4Chia Tai Tianqing Pharmaceutical Group Co., Ltd., Nanjing 210042, China; likun@cttq.com; 5Phase I Clinical Research Center, Huashan Hospital, Fudan University, Shanghai 200040, China

**Keywords:** doripenem, healthy Chinese subjects, prolonging infusion time, pharmacokinetics, Monte Carlo simulation

## Abstract

The aim of this study was to investigate the pharmacokinetics (PK) of doripenem in healthy Chinese subjects and evaluate the optimal dosage regimens of doripenem. A randomized, single-dose, three-period, self-crossover controlled extended-infusion clinical trial was conducted with 12 healthy Chinese subjects. Plasma and urine samples were collected to determine doripenem concentrations. Non-compartmental and population PK analysis were performed to characterize the PK of doripenem. The Monte Carlo simulation was employed to optimize dosing regimens based on the probability of target attainment of doripenem against pathogens with different minimum inhibitory concentrations (MIC). All 12 healthy Chinese subjects completed the study, and the doripenem was well tolerated. The study showed linearity relationships in the peak plasma concentration and the area under the concentration-time curve after intravenous infusion of doripenem from 0.25 g to 1.0 g. The cumulative urinary recovery rate of doripenem was 68.1–72.0% within 24 h. PPK modeling showed a two-compartmental model, with first-order elimination presenting the best fit for doripenem PK. Monte Carlo simulation results showed that 1.0 g q12h or 0.5 g q8h was an optimal regimen for pathogens susceptible to doripenem (MIC ≤ 1 mg/L); while high dose and extended infusion (1 g, q8h, 4 h infusion) was proposed for unsusceptible pathogens (2 ≤ MIC ≤ 8 mg/L). In the dose range of 0.25 to 1.0 g, doripenem showed linear pharmacokinetics. Doripenem at 1.0 g with a prolonged infusion time of 4 h was predicted to be effective against pathogens with MICs as high as 8 mg/L.

## 1. Introduction

Doripenem is an antibiotic drug in the carbapenem class [1]. It binds to bacterial penicillin-binding proteins to inhibit the synthesis of the bacterial cell wall, thus leading to cell death [2]. The drug has a broad spectrum and strong antibacterial activity against Gram-positive bacteria, Gram-negative bacteria, and anaerobic bacteria, and it is stable in the presence of most β-lactamases [3,4,5,6,7]. The in vitro antibacterial activity of doripenem is equivalent to that of imipenem or meropenem against Gram-positive bacteria and Gram-negative bacteria, respectively. Doripenem has the highest antibacterial activity against Pseudomonas aeruginosa, among all the other carbapenems tested in vitro [1,6,8]. A recent meta-analysis, including eight randomized controlled clinical trials, showed that doripenem in the treatment of acute bacterial infections was non-inferior to meropenem, imipenem, piperacillin, and tazobactam. The success rate in the treatment of complicated urinary tract infections (cUTIs) was higher than that of levofloxacin, ceftazidime, and avibactam [9]. At present, doripenem for injection has been marketed in Japan, the United States, and other countries, for the treatment of complicated abdominal infections and urinary tract infections [3,10]. In China, doripenem is still in the research and development stages.

The pharmacokinetic profiles of doripenem in healthy human subjects after intravenous infusion [11,12,13] showed that the drug reached a peak plasma concentration rapidly at the end of infusion, and the plasma protein binding rate was low (8.5%) [14]. The drug was widely distributed in various tissues in the body, and was metabolized by dehydropeptidase to generate the inactive main metabolite of M1, which accounts for about 18% of the original drug. The plasma elimination half-life (T_1/2_) of doripenem was short, about 1 h, and was mainly excreted through the kidney [15]. No drug accumulation was observed in blood or urine after multi-dose administration [12]. Doripenem is a time-dependent antibiotic; therefore, the most clinically relevant PK/PD index for doripenem efficacy is the percentage of the time during the dosing interval that the free drug concentration remains above the MIC (*f*T% > MIC). Increasing doripenem dosage or prolonging the infusion time could elevate the PK/PD index value and expect to achieve a better curative effect [3,16]. Based on human pharmacokinetic profiles and in vitro pharmacodynamic data, the Monte Carlo simulation was commonly used to evaluate the probability of reaching the PK/PD target following different dosing regimens. The simulation results could provide recommendations for the optimized dosage regimen in doripenem clinical applications [14,17]. Doripenem has not yet been marketed in China, and only limited pharmacokinetic studies were conducted in healthy Chinese subjects [18,19]. No published data are available on the safety and pharmacokinetic profiles after prolonged infusion time of doripenem and PK/PD analysis in healthy Chinese subjects. It is necessary to study whether the doripenem dosage regimens used abroad, especially the prolonged infusion time, are suitable for Chinese patients. This study aims to investigate the pharmacokinetic profiles of doripenem in healthy Chinese subjects after a single dose intravenous infusion for 1 h or 4 h, and aims to perform PK/PD analysis to provide information regarding the optimized dosage regimens of doripenem in Chinese patients.

## 2. Results

### 2.1. Demographics of Subjects and Safety

A total of 12 subjects, half of them male and half female, were enrolled in this study. All subjects completed the entire study following the protocol, and nobody withdrew from the trial. Therefore, the data obtained from all the 12 subjects entered the pharmacokinetic and safety analysis. The demographic statistics of the subjects are shown in Table 1.

All 12 participants enrolled in this study tolerated intravenous infusion of doripenem at 0.25 g, 0.5 g, and 1.0 g for 1 h, and 1.0 g for 4 h well, without severe drug-related adverse effects. Two mild drug-related adverse events were observed in the 0.5 g dose group during the study. One subject experienced headache and nausea, occurring at 5 h and 8 h after drug administration, respectively; the symptoms disappeared in the early morning the next day. Another subject showed an abnormal laboratory test of increased blood eosinophils of 9.3% (normal range is 0.4~8.0%) on the 6th day after drug administration; the laboratory test was normal on the 22nd day, in the follow-up visit. The incidence rate for both events was 8.3% (1/12).

### 2.2. Non-Compartmental Analysis of Plasma Concentrations

The average plasma concentration-time curves of doripenem after intravenous infusion are shown in Figure 1. The plasma PK parameters of doripenem are shown in Table 2. There was no significant difference in PK parameters observed between male and female subjects (*p* > 0.05); thus, the data were presented as the means and standard deviations from all 12 subjects.

### 2.3. Urinary Recovery

The bar charts of the average urinary recovery rate in each time segment, and the curves of cumulative urinary recovery rate time of doripenem after administration are shown in Figure 2 and Figure 3, respectively. Doripenem was mainly excreted through the kidneys after intravenous infusion; the accumulated urinary recovery rate during 0 to 24 h post-administration ranged from 68.1 to 72.0% with different doripenem doses and infusion times. The majority of doripenem was mainly excreted in urine within 0–4 h after 1 h intravenous infusion, and high doripenem concentration in urine was observed before 8 h post-administration (Table 3). The concentrations of doripenem in urine during the 4–8 h post-administration period were 119.4 ± 78.8 mg/L (range 32.6–334 mg/L) and 514 ± 266 mg/L (range 193–1140 mg/L) when the drug was infused for 1 h and 4 h, demonstrating that the extended infusion time increased doripenem concentration in urine by greater than fourfold.

### 2.4. Population Pharmacokinetic Analysis

A two-compartmental model with first-order elimination was the best fit for the data set. It showed the lowest OFV (OFV = 286), when compared to a one-compartmental model (OFV = 1311) and three-compartmental model (OFV = 299). Body weight was the only covariate on the peripheral volume of distribution (V_p_), when the OFV values decreased from 286 to 274. The final population PK parameters are summarized in Table 4. Goodness-of-fit plots for the final model are shown in Figure 4. The median of parameter estimates from a 1000-run bootstrap sets analysis are shown in Table 4. Each parameter estimate was within the range of the 95% confidence intervals, suggesting a robust final model. Based on the VPC plots (Figure 5), the final model can adequately predict the drug concentrations, with observed data being within the 90% prediction interval.

### 2.5. Pharmacokinetics/Pharmacodynamics

The probabilities of target attainment (PTAs) for different dosing regimens using the Monte Carlo simulations are shown in Figure 6. It shows that 1 h intravenous infusion with doripenem at 0.25 g every 12 h or 8 h could reach PTA ≥ 90% for pathogens with MIC ≤ 0.25 mg/L or 0.5 mg/L, respectively. As shown in Figure 5, 1 h intravenous infusion with doripenem at 0.5 g every 8 h could reach PTA of 99.8% for pathogens with MIC ≤ 1 mg/L. When the dose of doripenem increased to 1.0 g with 1 h infusion every 8 h, the calculated PTA reached 99.8% for pathogens with MIC ≤ 2 mg/L. With an extended infusion time of 4 h, doripenem at 1.0 g administered every 8 h resulted in PTA of 99.9% for pathogens with MIC ≤ 8 mg/L.

## 3. Discussion

In the present study, the PK parameters of doripenem obtained in healthy Chinese subjects were consistent with those reported in healthy human subjects in the United States and Japan, [11,12] in terms of T_1/2_ (1.06~1.15 h vs. 1.06~1.21 h), CL (12.1~14.5 L/h vs. 13.1~14.58 L/h), and V_d_ (19.9~23.1 L vs. 19.45~24.57 L). Although the mean values of CL and V_d_ in the 4 h infusion group were lower compared to the 1 h infusion group with 1 g doripenem administration, the differences were not statistically significant (*p* = 0.33 for CL, *p* = 0.61 for V_d_). The C_max_, AUC, and CL in the healthy Korean subjects [13] were slightly lower than those obtained in this study; however, the T_1/2_ and V_d_ values were similar. Comparing the PK profiles of doripenem plasma exposure obtained in this study with that in the healthy subjects in the United States [12], the C_max_ and AUC_0-inf_ after 4 h infusion with 1.0 g doripenem in the healthy Chinese subjects were slightly higher, without statistical significance. The C_max_ was 22.3 mg/L vs. 18.8 mg/L and the AUC_0-inf_ was 84.4 mg·h/L vs. 75.4 mg·h/L for Chinese vs. American, respectively. The slightly increased doripenem exposure in the Chinese subjects in comparison with the American subjects was due to the difference of body weight as a covariate influencing V_p_. Nevertheless, the difference has no clinical implications. The PK parameters of doripenem after 1 h infusion were similar to that reported in the two previous studies in healthy Chinese subjects [18,20]. In this study, extended infusion time of doripenem from 1 h to 4 h at the same dose (1.0 g) showed significant (*p* < 0.05) decreased C_max_ values. Other PK parameters such as T_1/2_, CL, CL_r_, and V_d_, were similar with different infusion times. The results of urinary recovery rate showed that the cumulative recovery rate during the 0 to 24 h period after intravenous infusion of doripenem at different doses was quite different among individual subjects, ranging from 43.8% to 80.8%. However, the average 0 to 24 h accumulative urinary recovery rates (68.1% to 72.0%) were not different in all the dose levels and regimens, and they were consistent with a published report (70.31~78.7%) [15,18].

In this study, a two-compartmental model with body weight as a covariate on V_p_ was developed to describe the PK characteristics of doripenem in healthy subjects. Typical population pharmacokinetic parameters were consistent with the previously published PPK models [14,21]. This result will contribute to the prediction of doripenem dosage regimens by PK/PD analysis and future studies in various patient populations. In patients, Ccr was regarded as the covariate in CL [16,19,21,22,23], but not in the healthy subjects, whose Ccr were in a narrow range (120 ± 17.3 mL/min) in the present study. Body weight was not found to be a covariate on CL and V_c_ in our model; however, we carried out simulations on V_p_ after body weight stratification with 40, 60, 80, and 120 kg. The PK/PD showed that dosing regimens did not need to change based on different body weights. One study reported that V_c_ and V_p_ with total BW as covariate in obese patients (BMI ≥ 40 kg/m^2^) and Monte Carlo simulations showed dosing regimens didn’t need to be adjusted in obese patients [23], which is consistent with our simulations.

Both in Japan and the US, optimal dosage regimens were provided based on underlying renal function by the population pharmacokinetic model and Monte-Carlo simulations. Similar PK parameters among healthy Chinese, Japanese, and American subjects indicate that the results of the pharmacokinetic/pharmacodynamic analysis in healthy subjects can be used as a reference for patients with normal renal function. In healthy subjects, the PK/PD target value of *f*%T > MIC was generally set at 35% or 40%. Our results and published data showed that following the standard doripenem dosing regimen (0.5 g once every 8 h infusion for 1 h) or prolonging the infusion time to 4 h, doripenem was effective against pathogens with MICs as high as 1 or 4 mg/L, respectively [16,17,19,22,23,24,25]. At present, the clinical indications of doripenem in Japan are septicemia, deep skin infections, UTIs, abdominal infections, etc. [11]. The approved indications of doripenem in the United States are for cUTIs and complicated abdominal infections [26]. Several clinical trials revealed doripenem had an excellent efficacy for complicated bacterial urinary tract infections [27,28,29,30]. In China, the current clinical research of doripenem focuses on cUTIs. A survey conducted by The China Antimicrobial Surveillance Network (CHINET) in 2018 showed that *Escherichia coli* (46.6%) was the main species isolated in urine samples, followed by *Klebsiella pneumonia* (10.0%), and *Pseudomonas aeruginosa* (3.7%) [31]. The in vitro study of doripenem susceptibility in 43 isolates of *Escherichia coli* obtained from Chinese patients with urinary tract infection showed that 95.3% of the isolates were very sensitive to doripenem with MIC < 0.06 mg/L. Thus, doripenem at 0.25 g infused for 1 h could achieve good microbiological efficacy for urinary tract infection caused by *Escherichia coli* in Chinese patients, based on the PK profile and PK/PD analysis. However, if the infection is caused by pathogens relatively insensitive to doripenem (MIC = 8 mg/L), extended infusion time to 4 h with a higher dose level could be considered as the treatment option. Additionally, in the treatment of adults with cUTIs, 0.5 g doripenem administered over 1 h every 8 h was recommended as a gold standard regimen in the United States and Europe [27,28,30]. In this study, the gold standard was adequate for pathogens with MICs as high as 1 mg/L, revealing that 0.5 g doripenem was also appropriate for Chinese patients with UTIs. This study is necessary to provide evidence for the dosing selection in Chinese patients.

The results from this study showed that the accumulated urinary recovery rate during the 0 to 24 h post-administration period ranged from 68.1 to 72.0% of the administered drug, confirming that doripenem was mainly excreted through the kidneys in healthy Chinese subjects. For patients with urinary tract infection, the main site of infection is in the urinary tract; therefore, the drug concentration in urine is a more relevant indicator than that in blood, for predicting the efficacy. The doripenem concentration detected in urine 4 h to 8 h post-administration following 0.25 g of doripenem was 31.7 ± 18.9 mg/L, which was about 3 times of C_max_ (11.9 ± 1.41 mg/L) in plasma and was much higher than the MIC value of Escherichia coli isolates (0.06 mg/L). The results from this study indicated that concentrations of doripenem at a dose range of 0.25 to 1.0 g infused for 1 h or 4 h in urine were high enough to cover MICs of the isolates, indicating that the regimens could be effective for Chinese patients with UTIs. However, due to the lack of other types of pathogenic bacteria that are responsible for urinary tract infection, and the low number of isolates of Escherichia coli tested, it is difficult to generalize the efficacy of the optimal dosage regimens of doripenem proposed in this study.

The occurrence of drug-related adverse events has been listed in the Japanese Finibax^®^ instructions [11]. The current study showed that doripenem, like other carbapenem antibiotics, is a safe antibiotic for use in healthy Chinese subjects, with few minor adverse events. The doripenem doses administered increased from 0.25 g to 1.0 g, and the prolonged infusion time of 4 h did not increase adverse events, which is consistent with published reports [12].

Several limitations existed in this study. Data from a small number of healthy subjects within a narrow range of demographic characteristics (age, weight, Ccr, etc.) was used for population PK and PK/PD analyses. Moreover, the optimal dosage regimens of doripenem were obtained using Monte Carlo simulations based on PK parameters from healthy volunteers. Since no data from patients was included in these analyses, any potential impact of infections on the PK of doripenem could not be assessed. Future clinical studies are warranted to confirm the efficacy and safety of the optimal dosage regimens for doripenem in infected patients, particularly in severe infection patients.

## 4. Materials and Methods

### 4.1. Study Design and Participants

This study was approved by the Ethics Committee of Huashan Hospital affiliated to Fudan University, and the clinical trial was registered at http://www.chinadrugtrials.org.cn (accessed on 15 January 2022) (registration number CTR20180182). Each subject signed an informed consent form before being enrolled in the study. In total, 12 healthy Chinese subjects were enrolled in the study from November 2017 to January 2018 at the Phase I Clinical Trial Center at Huashan Hospital. The study was conducted in two phases. The first phase was a randomized, open-label, three-period, self-crossover controlled trial. Doripenem was administered at a single dosage of 0.25 g, 0.5 g, and 1.0 g, with a 7-day washout time between each period. The intravenous infusion time was 1 h. The second phase was conducted in the same 12 subjects, to whom a 4 h intravenous infusion of doripenem at 1.0 g was administrated.

Twelve 18 to 45-year-old healthy Chinese subjects, with no abnormalities observed in the physical examination, 12-lead electrocardiogram, and clinical laboratory tests, nor in their vital signs, were enrolled in this study. Subjects with a history of allergies or severe allergic reactions to β-lactam antibiotics, a history of seizures or convulsions, a clear history of chronic diseases, pregnant and lactating women, and positive results in any of HIV-Ab, HBsAg, HCV-Ab, and syphilis tests were excluded.

### 4.2. Investigational Drug and Reference Standard Products

Doripenem for injection (batch number: 2010L05023, 0.25 g/bottle) and d5-doripenem (batch number: TQ-18201500887A, purity 98.14%) were provided by Chia Tai Tianqing Pharmaceutical Group Co., Ltd. (Nanjing, China). Doripenem reference standard (batch number: 1250-082A1, content 99.8%) was purchased from TLC Pharmaceutical Standards (Newmarket, ON, Canada).

### 4.3. Pharmacokinetic Sampling

Blood samples were collected before dosing, and at 15 min, 30 min, 45 min, 60 min, 75 min, 90 min, 105 min, 120 min, 180 min, 240 min, 360 min, 480 min, 600 min, and 720 min after the commencement of intravenous infusion, from all the subjects following each doripenem administration. About 4 mL of blood was collected in a lithium heparin anticoagulation tube. Plasma samples were prepared by centrifugation (4 °C, 3724 g, 10 min) and then stored at −70 ± 10 °C until analysis.

In the 1 h infusion group, urine samples were collected before and after the starting of intravenous infusion at 0~2 h, 2~4 h, 4~8 h, 8~12 h, and 12~24 h; in the 4 h infusion group, urine samples were collected before and after the starting of intravenous infusion at 0~4 h, 4~8 h, 8~12 h, and 12~24 h. The samples were also stored at −70 ± 10 °C until analysis.

### 4.4. Determination of Doripenem Concentration in Plasma and Urine Samples

A validated LC-MS/MS assay was used to determine the concentrations of doripenem in plasma and urine samples. The solid-phase extraction method was used to process the samples before analysis. The stable isotope d5-doripenem was used as the internal standard. The analytes were separated by liquid chromatography with ACQUITY UPLC^®^ BEH Phenyl column (2.1 mm × 50 mm; 1.7 µm) and eluted with gradient elution of methanol and 2 mM ammonium acetate as the mobile phases. An electrospray ionization source and an API 5500-QTRAP triple quadrupole mass spectrometer ((AB SCIEX, Framingham, MA, USA) with the positive ion mode and multiple-reaction monitoring were used for quantitative analysis of doripenem and its internal standard. The method validation results showed that the lower limit of quantification of doripenem was 0.100 mg/L for both plasma and urine samples. The doripenem standard curves ranged from 0.100 to 20.00 mg/L with good linearity (R^2^ > 0.99), and the ranges of quality control concentration were 0.300 to 16.0 mg/L in plasma and urine samples. The intra- and inter-assay accuracies were 95.2~112.8% and 93.0~111.6% for plasma and urine samples, respectively; the precisions were less than 5.2% and 9.8% for plasma and urine samples, respectively. No obvious matrix effects were detected. The stability tests showed that doripenem in plasma samples was stable at room temperature for 7 h, at −70 ± 10 °C for 114 days, and after 3 freeze–thaw cycles from −70 ± 10 °C to room temperature. The doripenem in whole blood samples was stable at room temperature for 1.5 h. The doripenem in urine samples was stable at room temperature for 7 h, at −70 ± 10 °C for 181 days, and after 3 freeze-thaw cycles from −70 ± 10 °C to room temperature.

### 4.5. Pharmacokinetic and Pharmacokinetic/Pharmacodynamic Analysis

The PK parameters of doripenem were calculated using the non-compartmental model with the Phoenix WinNonlin 8.0 (Pharsight corporation, Mountain View, CA, USA) software. The linearity of doripenem dose vs. AUC_0-last_, AUC_0-inf_ or C_max_ was analyzed by the Power model. The difference in PK parameters between males and females was analyzed by Student’s *t*-test and when the p-value less than 0.05 was considered statistically significant.

Population PK analysis was performed using NONMEM 7.4 (Icon Development Solutions, Ellicott City, MD, USA) with G77 FORTRAN compilers and FOCEI algorithm. The base model was selected among one-, two-, and three-compartment models, according to the objective function values and the goodness-of-fit plots. The tested covariates for total clearance (CL) were age, sex, body weight, creatinine clearance (Ccr) (determined by the Cockcroft-Gault equation), serum creatinine level (SCR), white blood cell count, total protein, albumin, serum alanine aminotransferase (ALT), and total bilirubin (TBIL); whereas the tested covariates for central volume of distribution (V_c_), peripheral volume of distribution (V_p_), and inter-compartment clearance (Q) were age, sex, body weight, white blood cell count, hemoglobin, total protein, and albumin. The bootstrap sampling and visual predictive checks (VPCs) were used to evaluate the robustness and predictive performance of the final model.

Dosing regimen evaluations were performed using Monte Carlo simulations. The target value of PK/PD index (*f*%T > MIC) was set as 35% [14] against Gram-negative bacteria (free fraction of doripenem used was 91.5% [25]). *f*%T > MIC were computed through 5000 simulations based on the parameter estimates and inter-individual variability from the final PPK model versus different MICs for each regimen. The PTAs for evaluated regimens were calculated by the percentage of *f*%T > MIC that achieved 35% against total simulations at different MICs.

## 5. Conclusions

In summary, all the regimens of doripenem administrated in this study were safe and well tolerated in healthy Chinese subjects. In the dose range of 0.25 g to 1.0 g, doripenem showed linear pharmacokinetic characteristics. Doripenem at 1.0 g, with an extended infusion time of 4 h, was predicted to be effective against pathogens with MICs as high as 8 mg/L by PK/PD analysis.

## Figures and Tables

**Figure 1 antibiotics-11-00958-f001:**
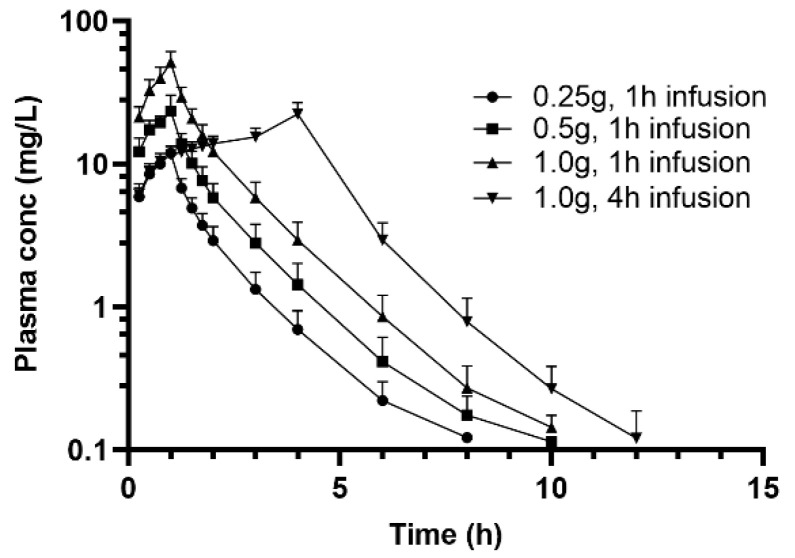
Mean plasma doripenem concentration-time curves after a single intravenous infusion of doripenem in healthy Chinese subjects. Symbols and error bars are the means and standard deviations from 12 subjects in each group.

**Figure 2 antibiotics-11-00958-f002:**
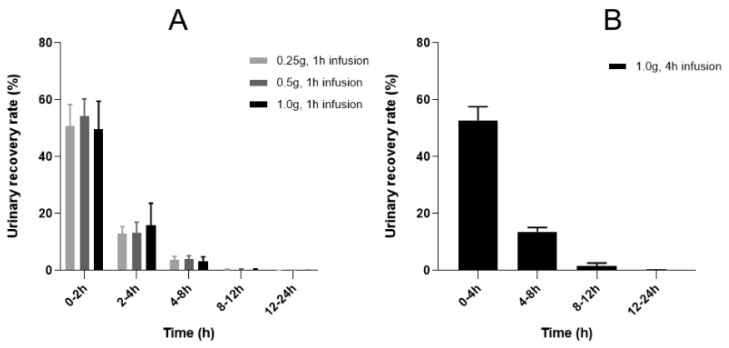
Mean segmented urinary recovery rates after a single intravenous infusion of doripenem in healthy subjects. Doripenem was infused for 1 h (**A**) or 4 h (**B**). Bars and error bars are the means and standard deviations from 12 subjects in each dose group.

**Figure 3 antibiotics-11-00958-f003:**
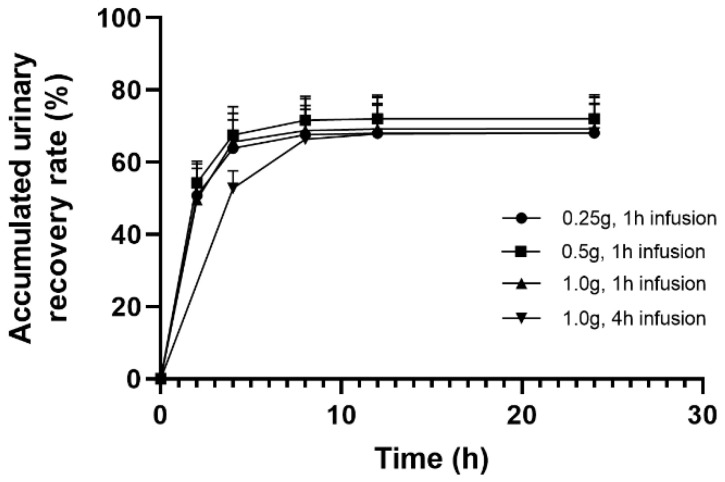
Mean accumulated urinary recovery rates after a single intravenous infusion of doripenem in healthy subjects. Symbols and error bars are the means and standard deviations from 12 subjects in each group.

**Figure 4 antibiotics-11-00958-f004:**
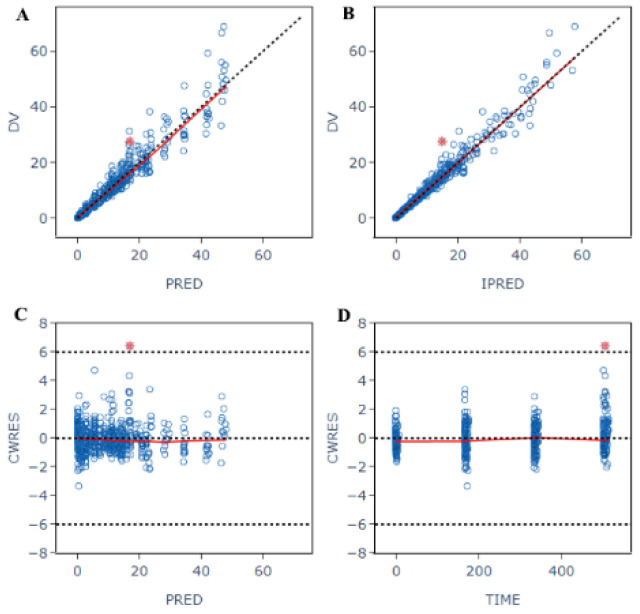
Goodness-of-fit plots for the final pharmacokinetic model. (**A**): Plot of observed concentrations versus population predictions. (**B**): Plot of observations versus individual predictions. (**C**): Conditional weighted residual versus population predictions. (**D**): Conditional weighted residual versus time. The open circles show observations. The red line shows smooth fitting for observations.

**Figure 5 antibiotics-11-00958-f005:**
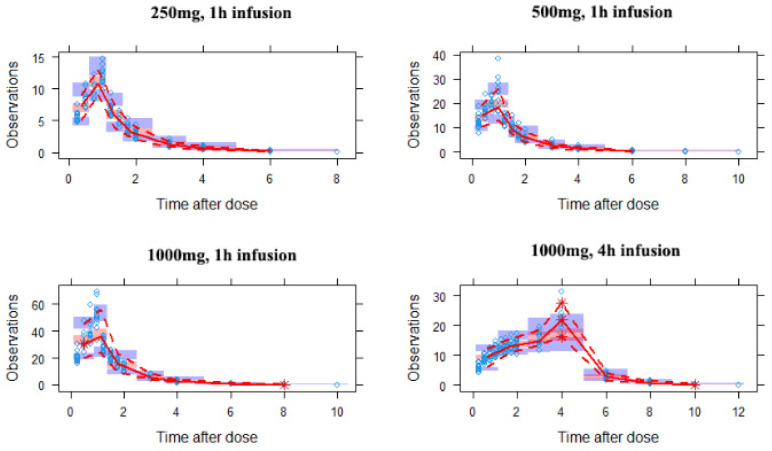
Visual predictive check plots for the doripenem concentration in plasma with different dosing regimens. Open circles represent observed concentrations. The red line represents the median of the observations. The red dashed lines represent the 5th and 95th percentiles of the observations. The blue shaded areas represent the 90% confidence intervals for the 5th and 95th percentiles, and the orange shaded areas represent the median of the predicted data.

**Figure 6 antibiotics-11-00958-f006:**
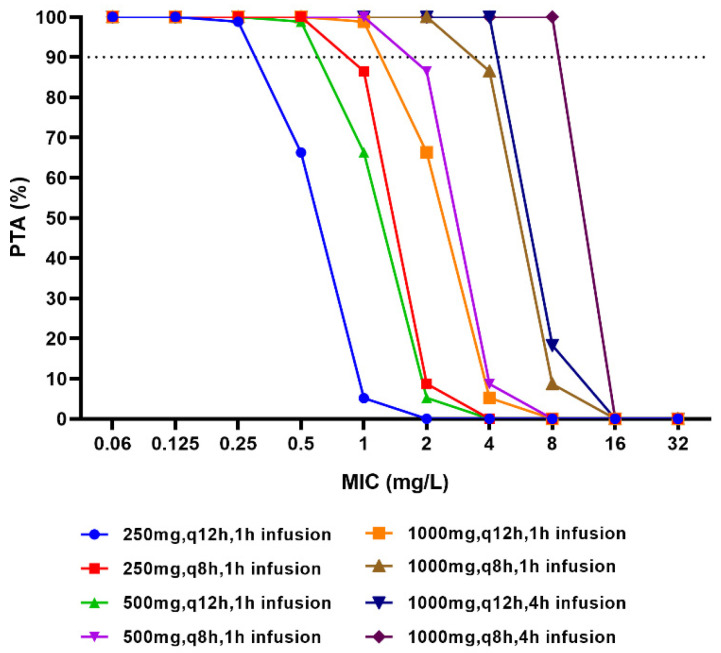
PTA results of doripenem dosing regimens. The lines show calculated PTA of doripenem in each simulated dosage regimen.

**Table 1 antibiotics-11-00958-t001:** Mean (SD) baseline demographic characteristics of healthy Chinese subjects (n = 12).

	Mean	SD
Gender	Male (n = 6), Female (n = 6)	
Age (years)	30.5	7.17
Height (cm)	164	9.59
Weight (kg)	61.3	9.71
BMI (kg/m^2^)	22.8	2.08
Ccr (mL/min)	120	17.3

Abbreviations: BMI = body mass index; Ccr = creatinine clearance rate; n = number of subjects; SD = standard deviation.

**Table 2 antibiotics-11-00958-t002:** Plasma pharmacokinetic parameters in healthy Chinese subjects after a single intravenous infusion of doripenem (n = 12).

Parameter	Unit	Doripenem Dose Group			
		0.25 g, 1 h Infusion	0.5 g, 1 h Infusion	1.0 g, 1 h Infusion	1.0 g, 4 h Infusion
C_max_	mg/L	11.9 (1.43)	23.9 (6.31)	51.3 (9.82)	22.3 (4.53)
T_max_	h	0.975 (0.0716)	0.932 (0.153)	0.993 (0.0132)	4 (0)
AUC_0–8 h_	h·mg/L	17.5 (2.95)	35.4 (7.18)	72.4 (13.1)	82.9 (12.1)
AUC_0–24 h_	h·mg/L	17.3 (3.02)	35.3 (7.38)	72.7 (13.3)	84.1 (12.5)
AUC_0-inf_	h·mg/L	17.6 (2.99)	35.6 (7.29)	72.9 (13.3)	84.4 (12.5)
AUC_ext_	%	2.13 (1.07)	0.989 (0.862)	0.405 (0.187)	0.338 (0.0739)
T_1/2_	h	1.06 (0.127)	1.10 (0.255)	1.15 (0.147)	1.14 (0.213)
MRT	h	1.04 (0.16)	1.05 (0.2)	1.1 (0.152)	1.23 (0.115)
CL	L/h	14.5 (2.26)	14.5 (2.60)	14.1 (2.38)	12.1 (1.76)
V_d_	L	22.1 (3.18)	22.8 (6.56)	23.1 (4.14)	19.9 (4.85)
CL_r_	L	9.87 (1.75)	10.4 (1.76)	9.87 (2.46)	8.21 (1.48)

Abbreviations: AUC = area under the concentration-time curve; CL = clearance; CL_r_ = renal clearance; C_max_ = peak concentration; g = gram; h = hour; L = liter; MRT = mean residence time; n = number of subjects; T_1/2_ = terminal half-time; T_max_ = time to reach peak concentration; SD = standard deviation; V_d_ = apparent volume of distribution. Note: data presented as mean (SD).

**Table 3 antibiotics-11-00958-t003:** Mean segmented doripenem concentration in urine of healthy Chinese subjects after a single intravenous infusion of doripenem (n = 12).

Time	Doripenem Concentration (mg/L)			
	0.25 g, 1 h Infusion	0.5 g, 1 h Infusion	1.0 g, 1 h Infusion	1.0 g, 4 h Infusion
0–2 h	667 (535)	974 (745)	2674 (1888)	1936 (996)
2–4 h	132 (57)	296 (202)	636 (339)	
4–8 h	31.7 (18.9)	59.2 (34.5)	119.4 (78.8)	514 (266)
8–12 h	3.34 (2.62)	4.05 (1.71)	10.1 (8.27)	34.9 (25.8)
12–24 h	0.304 (0.139)	0.676 (0.517)	1.27 (0.775)	2.73 (2.63)
0–24 h	ND	ND	ND	ND

Abbreviations: n = number of subjects; ND = not detectable. Note: data presented as mean (SD).

**Table 4 antibiotics-11-00958-t004:** Population pharmacokinetic parameters of doripenem.

Parameter	Estimate		Between-Subject Variability	
	Original Dataset (Typical Value)	Bootstrap Dataset (Median and 95% Interval Confidence)	Original Dataset (%)	Bootstrap Dataset (%)
CL (L/h)	14.2	14.2 (12.9, 15.5)	15.1	14.4
V_c_ (L)	8.17	8.20 (7.42, 8.67)	15.3	14.3
Q (L/h)	8.54	8.59 (7.30, 9.72)	0 (FIX)	\
V_p_ (L)	6.95	6.95 (6.24, 7.58)	7.37	6.70
θ_BW on_ _Vp_	0.713	0.700 (0.246, 0.987)	NA	NA
Proportional Error (%)	12.1	12.0 (10.4, 13.7)	NA	NA
Additive Error (%)	3.76	3.71 (2.59, 5.54)	NA	NA

Abbreviations: CL, clearance from central compartment; Q, intercompartmental clearance between central compartment and peripheral compartment; V_c_, central volume of distribution; V_p_, volume of distribution in peripheral compartment; NA, not applicable. θ_BW on Vp_, body weight was a covariate on volume of distribution in peripheral compartment. Of the bootstrap runs, 100% (1000/1000) converged successfully.

## Data Availability

The raw data supporting the conclusions of this article will be made available by the authors upon reasonable request.

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
