# Peer review of "Pharmacokinetics and Safety of Doripenem in Healthy Chinese Subjects and Monte Carlo Dosing Simulations"

_antibiotics, 2022, doi:10.3390/antibiotics11070958_

Round 1

Reviewer 1 Report

Wang Y. and Liu X. et al. is a well written manuscript describing the pharmacokinetics and safety of doripenem in 12 healthy chinese subjects as tested in a 3-period self-crossover-controlled plus one extended infusion clinical trial. The PK analysis was performed using non-compartmental analysis and population pharmacokinetics (PopPK). The PopPK estimated parameters were utilized for Monte Carlo simulations to further guide the optimized dosing regimen for doripenem for pathogens susceptible and unsusceptible to doripenem.

My comments are as follows:

1.      Line 127-128: Higher concentration of doripenem in urine for the 4-8hr collection window after the 4h infusion as compared to the 1h infusions is a result of the longer duration of infusion. As expected, given the short half-life of doripenem most of the drug after the 1h infusions was eliminated in urine in 0-2h as seen in Table 3. The longer infusion duration only delayed the elimination of doripenem in the urine and did not increase doripenem in the urine. So I would strongly recommend removing the part of the line 127 after the word ‘respectively’.

2.      For the NCA results in Table 2, please discuss why the CL and Vd were lower for the 4h infusion as compared to the 1h infusions.

3.      In section 4.5, please specify which covariates were tested on which PK parameter. Why was covariate testing done on peripheral volume? Given the small number of subjects (N=12) with extremely limited range of typical demographic characteristics, it is usually a better practice to only test covariates on CL and central volume.

4.      I would suggest discussing the impact of bodyweight (if any) on the CL and volume of distribution of doripenem as seen in the previously published PopPK models of doripenem

5.      For table 4, please specify in the footnote how many of the 1000 bootstrap runs converged successfully.

6.      In section 2.5, I recommend defining the acronym ‘PTA’ on first use and explaining how it was calculated for Figure 5.

7.      Study limitations: I suggest adding a paragraph for study limitations in the discussion section which includes (1) Data from a small number of healthy subjects within a narrow range of demographic characteristics (age, weight, Ccr, etc.) was used for these analyses. (2) Since no data from patients was included in these analyses, any potential impact of infections on the PK of doripenem could not be assessed.

8.      Line 91: I suggest to remove the words ‘were’ and ‘after’ from this sentence to fix the sentence structure and also apply this change in the abstract.

Reviewer 2 Report

Your paper on "Pharmacokinetics and safety of doripenem in ...." is interesting, well described and the study design is well conducted. It can be said that now we have the complete pharmacokinetics and safety of doripenem of all different type of population in the world.

Table 4; in the abbrevaiation doesn't appear  the description of all the parameters: what is  q BW on V2

In the same table: What does it mean Proportional? Proportional to what?

Also under the inscription Additive error (%) appears something written, please erase.

At page 5, "4.3 Pharmacokinetics sampling" it will be better to express the time in samples collection always in minutes, so 1.25 h is 85 min, 1.5 h  is 90 min etc. I think it's easier to follow the timing of samples collection.

In the same page, line 306 you write that intra- and inter-assay were 95.2-112.8% and 93.0-11.6% for plasma and urine samples, respectively. Are you sure that 11.6% is correct?

Reviewer 3 Report

A randomized, single-dose, 3-period self-crossover-controlled plus one extended infusion clinical trial was conducted on twelve healthy Chinese subjects. The study describes the results of the PK clinical trial. In the dose range of 0.25 g to 1.0 g, doripenem showed linear pharmacokinetic characteristics. The obtained results have therapeutic relevance, but the study lacks the research novelty. Nevertheless, I propose the publication after adding comparative literature results with the gold standard in the given indication.

Round 2

Reviewer 3 Report

The authors adequately addressed the referee's comments and modified the paper accordingly. Now the paper can be accepted for publication.